# BAM: BAYES WITH ADAPTIVE MEMORY

**Josue Nassar**[*]
Department of Electrical and Computer Engineering
Stony Brook University
josue.nassar@stonybrook.edu

**Jennifer Brennan**
Department of Computer Science
University of Washington
jrb@cs.washington.edu

**Ben Evans**
Department of Computer Science
New York University
benevans@nyu.edu

**Kendall Lowrey**[*]
Department of Computer Science
University of Washington
klowrey@cs.washington.edu

## ABSTRACT

Online learning via Bayes' theorem allows new data to be continuously integrated into an agent's current beliefs. However, a naive application of Bayesian methods in non-stationary environments leads to slow adaptation and results in state estimates that may converge confidently to the wrong parameter value. A common solution when learning in changing environments is to discard/downweight past data; however, this simple mechanism of "forgetting" fails to account for the fact that many real-world environments involve revisiting similar states. We propose a new framework, Bayes with Adaptive Memory (BAM), that takes advantage of past experience by allowing the agent to choose which past observations to remember and which to forget. We demonstrate that BAM generalizes many popular Bayesian update rules for non-stationary environments. Through a variety of experiments, we demonstrate the ability of BAM to continuously adapt in an ever-changing world.

## 1 INTRODUCTION

The ability of an agent to continuously modulate its belief while interacting with a non-stationary environment is a hallmark of intelligence and has garnered a lot of attention in recent years (Zhang et al., 2020; Ebrahimi et al., 2020; Xie et al., 2020). The Bayesian framework enables online learning by providing a principled way to incorporate new observations into an agent's model of the world (Jaynes, 2003; Gelman et al., 2013). Through the use of Bayes' theorem, the agent can combine its own (subjective) *a priori* knowledge with data to achieve an updated belief encoded by the posterior distribution.

The Bayesian framework is a particularly appealing option for online learning because Bayes' theorem is closed under recursion, enabling continuous updates in what is commonly referred to as the recursive Bayes method (Wakefield, 2013). As an example, suppose the agent first observes a batch of data, $\mathcal{D}_1$, and then later observes another batch of data, $\mathcal{D}_2$. We can express the agent's posterior distribution over the world, where the world is represented by $\theta$, as

$$p(\theta|\mathcal{D}_1, \mathcal{D}_2) = \frac{p(D_2|\theta)p(\theta|\mathcal{D}_1)}{p(\mathcal{D}_2|\mathcal{D}_1)}, \tag{1}$$

where

$$p(\mathcal{D}_2|\mathcal{D}_1) = \int p(\mathcal{D}_2|\theta)p(\theta|\mathcal{D}_1)d\theta. \tag{2}$$

Equation 1 demonstrates the elegance and simplicity of recursive Bayes: at time $t$, the agent recycles its previous posterior, $p(\theta|\mathcal{D}_{<t})$, where $\mathcal{D}_{<t} = \{\mathcal{D}_1, \cdots, \mathcal{D}_{t-1}\}$, into its current prior and then combines it with a newly observed batch of data, $\mathcal{D}_t$, to obtain an updated posterior, $p(\theta|\mathcal{D}_{\leq t})$.

---

[*]corresponding authors

At first glance, it would appear that a naive application of recursive Bayes would suffice for most online learning tasks. However, the recursive Bayes method relies on the assumption that the world is stationary, i.e. $\mathcal{D}_1, \mathcal{D}_2, \cdots$ are all independent and identically distributed. When this assumption is violated, recursive Bayes can fail catastrophically. As an illustration, consider the law of total variance:

$$\text{Var}(\theta|\mathcal{D}_{<t}) = \mathbb{E}[\text{Var}(\theta|\mathcal{D}_{<t}, \mathcal{D}_t)\big|\mathcal{D}_{<t}] + \text{Var}(\mathbb{E}[\theta|\mathcal{D}_{<t}, \mathcal{D}_t]\big|\mathcal{D}_{<t}). \tag{3}$$

Since both terms on the right hand side are positive, equation 3 reveals that in expectation, the variance of the posterior decreases as more data is seen **regardless of the actual distribution of** $\mathcal{D}_t$, i.e.

$$\text{Var}(\theta|\mathcal{D}_{<t}) \geq \mathbb{E}[\text{Var}(\theta|\mathcal{D}_{<t}, \mathcal{D}_t)\big|\mathcal{D}_{<t}]. \tag{4}$$

In fact, for some models equation 4 is true with probability 1; we demonstrate examples in Appendix A. Thus, if the parameters of the environment, $\theta$, were to change, the variance of the posterior would still decrease, becoming more certain of a potentially obsolete parameter estimate. Modeling the environment as stationary when it is actually changing also keeps the learning speed of the agent artificially low, as tighter posteriors prevent large jumps in learning. This is the opposite of what an intelligent agent should do in such an event: if the environment changes, we would expect the agent's uncertainty and learning speed to increase in response.

As was elegantly stated by Monton (2002), the problem with naive use of recursive Bayes is that "Such a Bayesian never forgets." Previous approaches on enabling recursive Bayes to work in non-stationary settings have primarily focused on *forgetting* past experience either through the use of changepoint detection (Adams & MacKay, 2007; Li et al., 2021), or by exponentially weighting past experiences (Moens, 2018; Moens & Zénon, 2019; Masegosa et al., 2020). While empirically successful, their focus on forgetting the past means that revisited states are treated as novel. In this work we take an alternative approach to online Bayesian learning in non-stationary environments by endowing an agent with an explicit memory module. Crucially, the addition of a memory buffer equips the agent with the ability to modulate its uncertainty by choosing what past experiences to both forget and remember. We call our approach Bayes with Adaptive Memory (BAM) and demonstrate its wide applicability and effectiveness on a number of non-stationary learning tasks.

## 2 BAYES WITH ADAPTIVE MEMORY

The generative model is assumed to evolve according to

$$\theta_t \sim p(\theta_t|\theta_{t-1}, t), \tag{5}$$
$$\mathcal{D}_t \sim p_t(\mathcal{D}) \equiv p(\mathcal{D}|\theta_t), \tag{6}$$

where equation 5 is the latent dynamics that dictate the evolution of the environment parameters, $\theta_t$, and equation 6 is the likelihood whose parametric form is fixed throughout time, i.e. $p_t(\mathcal{D}) = \mathcal{N}(\theta_t, \sigma^2)$. Equations 5 and 6 define a state-space model, which allows one to infer $\theta_t$ through Bayesian filtering (Särkkä, 2013)

$$p(\theta_t|\mathcal{D}_{\leq t}) \propto p(\mathcal{D}_t|\theta_t)p(\theta_t|\mathcal{D}_{<t}), \tag{7}$$

$$p(\theta_t|\mathcal{D}_{<t}) = \int p(\theta_t|\theta_{t-1}, t)p(\theta_{t-1}|\mathcal{D}_{<t})d\theta_{t-1}. \tag{8}$$

The parameterization of equations 5 and 6 dictate the tractability of equations 7 and 8. If a priori an agent knew that equation 5 is a linear dynamical system with additive white Gaussian noise and equation 6 is also Gaussian whose conditional mean is a linear function of $\theta_t$, then the Kalman filter can be used (Kalman, 1960). For more complicated latent dynamics and/or likelihood models, methods such as particle filtering (Doucet & Johansen, 2009) and unscented Kalman filtering (Julier & Uhlmann, 1997) can be used. Crucially, Bayesian filtering methods assume that the latent dynamics governed by equation 5 are known; however, this is rarely the case in practice. Instead of making assumptions on the parametric form of equation 5, we take a different approach.

In BAM, the agent maintains a memory buffer, $\mathcal{D}_{<t}$, that stores previous observations of the environment. At time $t$ the agent obtains a new batch of data, $\mathcal{D}_t \sim p_t(\mathcal{D})$. How should the agent combine the newly observed data, $\mathcal{D}_t$, with its stored memory, $\mathcal{D}_{<t}$, to update its belief as encoded by the posterior distribution?

In recursive Bayes, the posterior distribution is computed according to[1]

$$p(\theta_t|\mathcal{D}_t, \mathcal{D}_{<t}) \propto p(\mathcal{D}_t|\theta_t)p(\theta_t|\mathcal{D}_{<t}), \tag{9}$$

$$p(\theta_t|\mathcal{D}_{<t}) \propto p(\theta_t) \prod_{j=1}^{t-1} p(\mathcal{D}_j|\theta_t), \tag{10}$$

where we refer to $p(\theta_t)$ as the base prior. Equation 10 allows us to interpret recursive Bayes as the agent constructing a dynamic prior, $p(\theta_t|\mathcal{D}_{<t})$, using all the experiences stored in its memory buffer. This works under the stationarity assumption; when this assumption is violated, the application of Bayes' theorem can lead to **confidently wrong results** as the "distance" between $p_i(\mathcal{D})$ and $p_j(\mathcal{D})$ can be vast. An alternative is for the agent to completely forget all of its past experiences

$$p(\theta_t|\mathcal{D}_t) \propto p(\mathcal{D}_t|\theta_t)p(\theta_t). \tag{11}$$

While equation 11 may be viable in situations where $\mathcal{D}_t$ is sufficiently informative, it is wasteful when experiences in the memory buffer may help infer $\theta_t$.

BAM dynamically finds a middle ground between these two extremes of remembering (equation 10) and forgetting (equation 11) everything by allowing the agent to choose which data to use from its memory buffer to construct the prior. Specifically, the agent is endowed with a time-dependent readout weight, $W_t = [w_{t,1}, w_{t,2}, \cdots, w_{t,t-1}]$ where $w_{t,j} \in [0, 1]$. Given a new datum $\mathcal{D}_t$, BAM constructs its posterior according to

$$p(\theta_t|\mathcal{D}_t, \mathcal{D}_{<t}, W_t) \propto p(\theta_t)p(\mathcal{D}_t|\theta_t) \prod_{j=1}^{t-1} p(\mathcal{D}_j|\theta_t)^{w_{t,j}}. \tag{12}$$

We can rewrite equation 12 as

$$p(\theta_t|\mathcal{D}_t, \mathcal{D}_{<t}, W_t) = \frac{p(\mathcal{D}_t|\theta_t)p(\theta_t|\mathcal{D}_{<t}, W_t)}{p(\mathcal{D}_t|\mathcal{D}_{<t}, W_t)}, \tag{13}$$

where

$$p(\theta_t|\mathcal{D}_{<t}, W_t) \propto p(\theta_t) \prod_{j=1}^{t-1} p(\mathcal{D}_j|\theta_t)^{w_{t,j}}, \tag{14}$$

and

$$p(\mathcal{D}_t|\mathcal{D}_{<t}, W_t) = \int p(\mathcal{D}_t|\theta_t)p(\theta_t|\mathcal{D}_{<t}, W_t)d\theta_t. \tag{15}$$

The prior construction in equation 14 is akin to recursive Bayes, but now the agent can dynamically and adaptively change its prior by using the readout weights, $W_t$, to weigh the importance of previous experience where at the extreme, it can choose to completely forget a previous experience, $w_{t,j} = 0$, or fully remember it, $w_{t,j} = 1$. For simplicity, we restrict the readout weights to be binary, i.e. $w_{t,j} \in \{0, 1\}$.

The combination of a memory buffer, $\mathcal{D}_{<t}$, with a time-dependent readout weight, $W_t$, allows BAM to generalize many previously proposed approaches. By setting $w_{t,1} = w_{t,2} = \cdots = w_{t,t-1} = 1$, we recover recursive Bayes (equation 10). By setting $w_{t,1} = w_{t,2} = \cdots = w_{t,t-1} = \alpha$, where $0 \le \alpha \le 1$ we recover the power priors approach of Ibrahim et al. (2015). By setting $w_{t,j} = \alpha^{t-1-j}$, where $0 \le \alpha \le 1$, we recover exponential forgetting (Moens, 2018; Moens & Zénon, 2019; Masegosa et al., 2020). Lastly, by setting a particular subset of the readout weights to be 0, we recover Bayesian unlearning (Nguyen et al., 2020).

The ability to adaptively change its prior implies that BAM can increase/decrease its uncertainty as the situation demands; subsequently, this modulates the agent's learning speed. Using variance as a proxy for uncertainty, one would expect that the variance of the prior used in BAM (equation 14) is always at least as large as the variance of the prior used in recursive Bayes (equation 10). We formalize this for the case of binary readout weights in the following proposition.

**Proposition 1.** *Let $p(\theta|\mathcal{D}_{<t}, W_t)$ be the prior used by BAM, defined in equation 14 and let $p(\theta|\mathcal{D}_{<t})$ be the recursive Bayes prior, defined in equation 13. Then*

$$\mathbb{E}\big[Var(\theta|\mathcal{D}_{<t}, W_t)\big|W_t\big] \ge \mathbb{E}[Var(\theta|\mathcal{D}_{<t})], \quad \forall W_t \in \{0, 1\}^{t-1}. \tag{16}$$

*Proof.* Proof is in Appendix B. □

---

[1] Recursive Bayes is equivalent to Bayesian filtering when $p(\theta_t|\theta_{t-1}, t) = \delta(\theta_t = \theta_{t-1})$.

## 2.1 Selection of Readout Weights via Bayesian Model-selection

While the previous section demonstrated the flexibility of BAM, the question remains: how should the readout weights, $W_t$, be set? Equation 13 allows us to view different readout weights as different models. Through this lens, we can follow the spirit of Bayesian model selection (Gelman et al., 2013) and compute a posterior over the readout weights

$$p(W_t|\mathcal{D}_t, \mathcal{D}_{<t}) \propto p(W_t|\mathcal{D}_{<t})p(\mathcal{D}_t|W_t, \mathcal{D}_{<t}). \tag{17}$$

For practicality, we compute the maximum a posteriori (MAP) estimate of equation 17 (Gelman et al., 2013) and use that as the value of the readout weight

$$W_t = \underset{W \in \{0,1\}^{t-1}}{\mathrm{argmax}} \ \log p(\mathcal{D}_t|W, \mathcal{D}_{<t}) + \log p(W|\mathcal{D}_{<t}), \tag{18}$$

$$= \underset{W \in \{0,1\}^{t-1}}{\mathrm{argmax}} \ \log \int p(\mathcal{D}_t|\theta_t)p(\theta_t|W, \mathcal{D}_{<t})d\theta_t + \log p(W|\mathcal{D}_{<t}). \tag{19}$$

The first term of equation 18 is the log marginal likelihood, which measures the likelihood of $\mathcal{D}_t$ being distributed according to the predictive distribution, $p(\mathcal{D}|W, \mathcal{D}_{<t})$ while the prior, $\log p(W|\mathcal{D}_{<t})$, acts as a regularizer. This procedure of constantly updating the readout weights through equation 18 can be interpreted as providing Bayes a feedback mechanism: equation 18 allows the agent to directly measure its ability to fit the observed data using different combination of experiences in its buffer via the readout weight, and then choosing the readout weight that leads to best fit. In contrast, standard Bayesian inference is an open-loop procedure: data, likelihood and prior are given and a posterior is spat out, irrespective of the fit of the model to the data (Simpson et al., 2017).

Still left is the question of how do we design the prior, $p(W|\mathcal{D}_{<t})$. In certain scenarios, using an uninformative prior, i.e. $p(W|\mathcal{D}_{<t}) \propto 1$, may suffice if the data is very informative and/or the number of data points in $\mathcal{D}_t$ is large. In scenarios where these conditions are not met, it is important to use an informative prior as it reduces the chance of overfitting. In general, the design of priors is highly nontrivial (Winkler, 1967; Gelman et al., 2013; Simpson et al., 2017). While there exists many potential options, we use penalized model complexity priors proposed by Simpson et al. (2017) as they are designed to reduce the chance of overfitting. Following Simpson et al. (2017), we parameterize the prior as

$$p(W|\mathcal{D}_{<t}) \propto \exp\left(-\lambda\sqrt{2\mathbb{D}_{KL}[p(\theta_t|W, \mathcal{D}_{<t})\|p(\theta_t)]}\right), \tag{20}$$

where $\lambda \in [0, \infty)$ is a hyperparameter that controls the strength of the prior.[2] Equation 20 encodes our prior belief that we favor values of $W_t$ that produce simpler models, where simplicity is quantified as the Kullback-Leibler divergence between $p(\theta_t|W_t, \mathcal{D}_{<t})$ and the base prior, $p(\theta_t)$.

Plugging equation 20 into equation 18 we get

$$W_t = \underset{W \in \{0,1\}^{t-1}}{\mathrm{argmax}} \ \log p(\mathcal{D}_t|W, \mathcal{D}_{<t}) - \lambda\sqrt{2\mathbb{D}_{KL}[p(\theta_t|W, \mathcal{D}_{<t})\|p(\theta_t)]}. \tag{21}$$

In general, solving equation 21 is difficult as the number of possible readout weights is $2^{(t-1)}$, making brute force solutions practically infeasible. While there exists many approaches for performing discrete optimization, we found that using a simple greedy approach sufficed for our experiments; in the interest of space, we defer discussion regarding this to Appendix C.

## 3 Related Works

A variety of approaches have been proposed for learning in non-stationary environments. In signal processing, adaptive filtering techniques such as recursive least squares (RLS) and least mean square filtering (LMS) are the de facto approaches for filtering in non-stationary environments (Haykin, 2008). While empirically successful, RLS and LMS are only applicable for a limited range of models, i.e. linear models. In contrast, BAM is a general purpose algorithm that can be deployed on a wide variety of models.

---

[2]$\lambda = 0$ recovers the uninformative prior case, $p(W_t|\mathcal{D}_{<t}) \propto 1$.

If the latent dynamics are known—or assumed to be known—then Bayesian filtering can be employed. A popular approach is to model the latent dynamics (equation 5) as an autoregressive process (Kurle et al., 2020; Rimella & Whiteley, 2020). While this approach has been popular, it is only applicable for models where the parameters are real-valued. A seminal work on Bayesian filtering is the Bayesian online changepoint detction (BOCD) algorithm of Adams & MacKay (2007), where the latent dynamics (equation 5) are modeled to be piece-wise constant. While BOCD is broadly applicable and has seen empirical success, the caveat is that an agent forgets all previous experience when a change is detected; thus, previously visited states appear novel to the agent and learning must begin from scratch. An extension to BOCD was proposed by Li et al. (2021), where when a change is detected a scaled version of the previous posterior is used as the prior. While similar in spirit to BAM, we note that the approach proposed in Li et al. (2021) is designed for Gaussian distributions, while BAM can work with arbitrary distributions. Moreover, the approach in Li et al. (2021) can only increase the uncertainty by a **fixed pre-determined amount** while BAM can adaptively modulate its uncertainty.

Previous works have proposed solutions for making recursive Bayes more suited for use in non-stationary environments through exponential forgetting of past data (Moens, 2018; Moens & Zénon, 2019; Masegosa et al., 2020). While these models have also seen empirical success, their focus have been on forgetting past experiences which prevents the agent to leverage past experiences that are relevant. In BAM, the agent is focused not only on forgetting irrelevant experiences **but remembering relevant experiences as well**.

The use of readout weights in BAM can be seen as an instance of likelihood tempering, which has been used to perform robust Bayesian inference (Wang et al., 2017) and to help with approximate Bayesian inference schemes (Neal, 1996; 2001; Mandt et al., 2016). While previous works focus on the offline case where data has already been collected, BAM focuses on the online case where the agent adaptively tempers the likelihood.

The concept of an external memory buffer has recently been explored in machine learning (Gemici et al., 2017; Wu et al., 2018; Marblestone et al., 2020). While similar in spirit to BAM, most works use a softmax as their readout weight. As a byproduct, the agent **must** select an element from the buffer even if it isn't applicable to the task at hand! BAM has no such restriction, and can ignore all the previous data in the buffer, resetting back to the base prior.

## 4 EXPERIMENTS

To demonstrate the versatility of BAM, we apply it in a variety of scenarios. As BAM is a learning paradigm, it can be implemented as a module in a larger framework allowing it to be easily used in settings such as control/reinforcement learning and domain adaptation (Thompson, 1933; Osband et al., 2018; Lowrey et al., 2018; Yoon et al., 2018). BAM requires the ability to construct the posterior, $p(\theta_t|\mathcal{D}_{<t}, W_t)$, and evaluate the log marginal likelihood, $\log p(\mathcal{D}_t|\mathcal{D}_{<t}, W_t)$. In general, the posterior and log marginal likelihood are only available analytically for conjugate priors (Gelman et al., 2013). While approaches exist for approximating the posterior (Robert et al., 2004; Brooks et al., 2011; Blei et al., 2017) and the log marginal likelihood (Robert et al., 2004; Gelman et al., 2013; Grosse et al., 2015), we restrict ourselves to only use conjugate priors to ensure any benefits of BAM are not due to uncertain effects of approximations. The use of conjugate priors also allows us to use sufficient statistics to compute posteriors, allowing BAM to scale amicably when the number of data points in a batch is large (Casella & Berger, 2021).

### 4.1 EXPERIMENT 1: INFERENCE IN A NON-STATIONARY ENVIRONMENT

To evaluate BAM on online inference in a non-stationary environment, we generate data from the following model

$$\theta_t = a\sin\left(\frac{2\pi t}{100}\right) + b, \tag{22}$$

$$p(\mathcal{D}_t|\theta_t) = \text{Binomial}(15, \theta_t), \tag{23}$$

where $a = 0.3$ and $b = 0.5$ are chosen such that the lower and upper bounds for $\theta_t$ are 0.2 and 0.8, respectively. We evaluate BAM with no regularization, $\lambda = 0$, and with regularization, where $\lambda = 0.1$; as the data is discrete, there is a possibility that BAM could overfit, thus a priori we

would expect the regularized BAM to perform better. We compare against recursive Bayes, Bayesian exponential forgetting (BF) and Bayesian online changepoint detection (BOCD). [3]

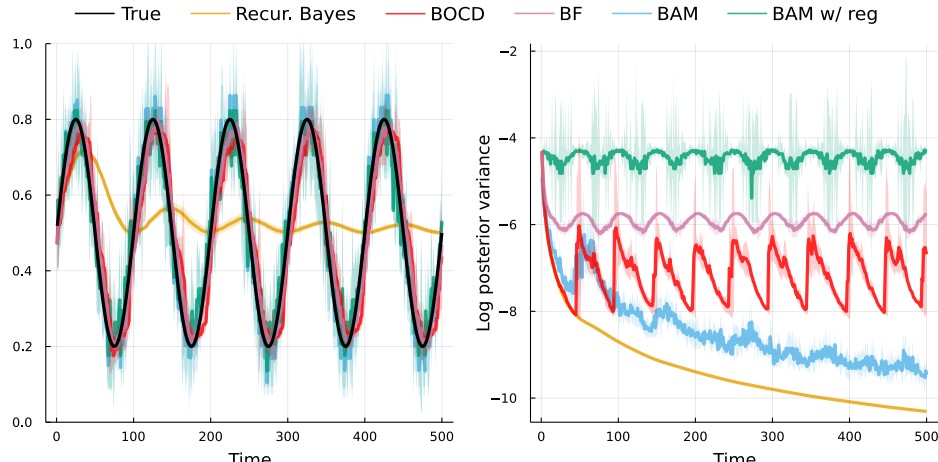

Figure 1: Comparison of recursive Bayes, BOCD, BF, BAM and BAM with regularization on inferring a time-varying parameter. Results are computed over 20 random seeds. Solids lines are the median and and error bars represent the 10% and 90% quantiles. **Left**: Temporal evolution of the posterior mean. **Right**: Temporal evolution of the log posterior variance.

Figure 1 demonstrates the weakness of recursive Bayes; as it views more data, the posterior gets more confident. This reduces the learning speed of the agent, preventing it from accurately tracking $\theta_t$, and causing it to converge to the average with an extremely low posterior variance. BOCD tracks the parameter relatively well, though its estimates are slightly delayed. As BOCD lacks the ability to recall useful data from the past, its posterior variance resets every time a changepoint is detected. BAM is able to track $\theta_t$ and doesn't suffer from temporal lag seen in the BOCD results, though the lack of regularization leads to estimates that are not as smooth as BOCD. The posterior variance of BAM reflects that the agent remembers relevant history and forgets irrelevant history, reducing the jump in posterior variance when revisiting a previously seen state. Lastly, we can see that BAM with regularization leads to smoother estimates but tends to be less confident compared to the other methods.

## 4.2 EXPERIMENT 2: CONTROLS

In this section we illustrate the benefit of memory by applying BAM on a learning task to model non-linear dynamics for controls. The task is an analytical version of Cartpole (Barto et al., 1983), where the goal is to swing-up a pole on a moving cart. Non-stationarity is introduced by changing the environment's gravity over time. We explore the performance of BAM under two different information models. In the episodic setting, the agent is told when a change occurs, but not the value of the new gravity parameter. In the continual learning setting, the agent is not informed of environmental changes.[4]

The reward for the task is the cosine of the angle of the pole on the cart, where an angle of $0°$ is the vertical 'up' position. To make the problem amenable for analytical posterior and log marginal likelihood computations, we model the nonlinear dynamics using linear regression with random Fourier features (RFF) (Rahimi & Recht, 2007)

$$x_t = x_{t-1} + M\phi(x_{t-1}, a_t) + \varepsilon_t, \quad \varepsilon_t \sim \mathcal{N}(0, \sigma^2 I), \tag{24}$$

---

[3]The timescale parameter for BOCD is 1/100, which is the frequency of the sinusoid. The weighting term for BF is 0.8.

[4]For both settings, the number of data points in a batch is relatively large, leading the log marginal likelihood to overtake the prior in equation 21. As regularization has little effect, results are shown for $\lambda = 0$.

where $x_t \in \mathbb{R}^{d_x}$ is the state vector, $a_t \in \mathbb{R}^{d_a}$ is the action vector, $\varepsilon_t \in \mathbb{R}^{d_x}$ is state noise and $\phi$ is our RFF function. For simplicity, we assume a fixed noise variance of $\sigma^2 = 10^{-6}$. This parameterization allows us to perform Bayesian linear regression over $M$ which is analytically tractable (Gelman et al., 2013). Full details can be found in Appendix D.1.

### 4.2.1 EPISODIC ONE-SHOT

In this setting our simulated Cartpole hypothetically moves between different planets—hence a change in gravity—while still being asked to swing the pole up. In an episode, gravity is fixed for the duration of 15 trials, where each trial resets the Cartpole to a random initial state, $x_0$. Each trial produces a trajectory of states and actions of length $H$ that are batched into one unit of data, such that each episode contributes 15 experiences; thus the datum for trial $t$ is $\mathcal{D}_t = \{([x_j, a_j], [x_j - x_{j-1}])\}_{j=1}^{H}$.

We compare BAM to recursive Bayes in a one-shot manner: after the first trial of a new episode, BAM computes a weight vector over all previously encountered trial data to inform a posterior for the duration of the episode. Recursive Bayes is reset to the base prior at the beginning of a new episode. Both proceed to update their belief every trial in an episode.

We show in Figure 2 results over 5 random seeds where the expected score for a ground truth model is shown as a reference. The first time BAM encounters a novel planet, it resets its prior to the base prior and starts learning from scratch, similar to recursive Bayes. On subsequent visits however, BAM is able to leverage its past experiences to quickly adapt and recover high levels of performance. As recursive Bayes starts from scratch, it will again need multiple trials to develop a competent model.

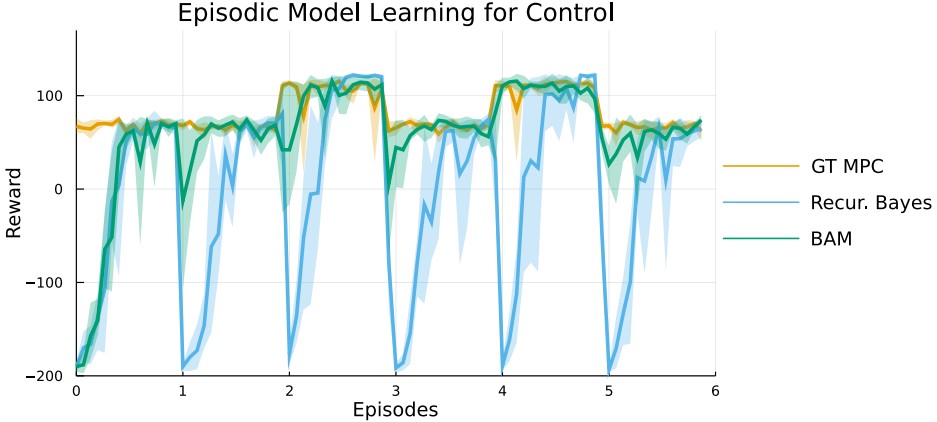

Figure 2: We compare BAM to recursive Bayes in an episodic model learning for control setting where the agent knows when the cartpole dynamics have changed. For reference, the ground-truth scores (GT MPC) are also plotted for the different settings; all methods were averaged across 5 random seeds with 10-90% quantiles shown. In this sequence, the values of gravity per episode are 9.81, 11.15, 3.72, 11.15, 3.72, 9.81.

### 4.2.2 CONTINUAL LEARNING

In addition to the challenge of adapting to a different environment, we also test BAM when the agent is not informed of the change, such that adaption must happen continually. In this scenario without explicit episodes, the gravity of the world can change after a trial, unbeknownst to the agent. Similar to the previous setting, a datum for trial $t$ is $\mathcal{D}_t = \{([x_j, a_j], [x_j - x_{j-1}])\}_{j=1}^{H}$.

While it is straightforward to run BAM in this setting, we also investigate combining BAM with BOCD, which we denote as BAM + BOCD. In BOCD, the detection of a changepoint causes the posterior distribution to be reset to the base prior. In BAM + BOCD, the detection of a changepoint is used as signal for when the agent should adapt its prior by computing $W_t$, to obtain $p(\theta_t | W_t, \mathcal{D}_{<t})$; this avoids rerunning the optimization procedure after each trial.

We show in Figure 3 that while BOCD works as intended, without BAM the Cartpole has to relearn a model from the prior belief, leading to significant dips in the expected reward. While all methods are

able to adapt when the environment is in a constant state, the use of past information allows BAM and BAM + BOCD to quickly adapt. We can see that BAM and BAM + BOCD perform very similarly to each other, suggesting that we can bypass unnecessary computation.

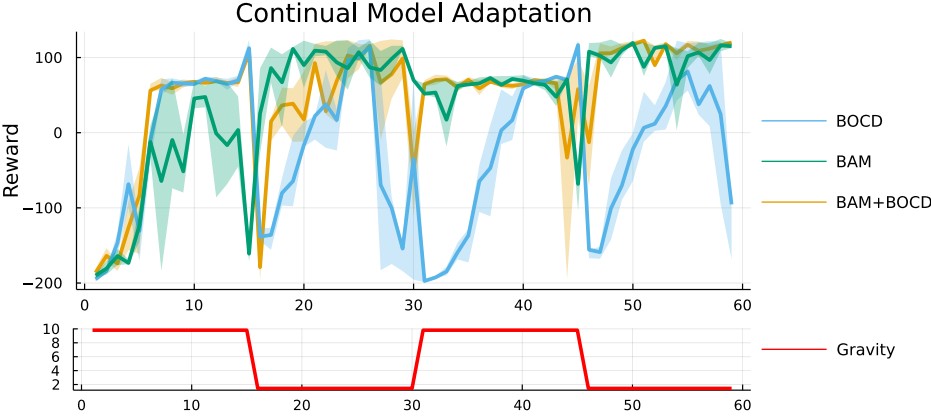

Figure 3: The bottom plot shows the gravity value changing over time; the performance of the learned model in the swing-up task is shown above for 5 random seeds with 10-90% quantiles. Initially all methods perform similarly until BAM encounters an environment it has experienced before, in which it adapts more quickly than BOCD. The initial similarity is due to both methods detecting similar changepoints and reverting to un-informed priors. Naive BAM performs well, albeit with weight optimization after each trail versus only during detected changepoints for the other methods (60 optimizations versus ~5).

### 4.3 EXPERIMENT 3: NON-STATIONARY BANDIT

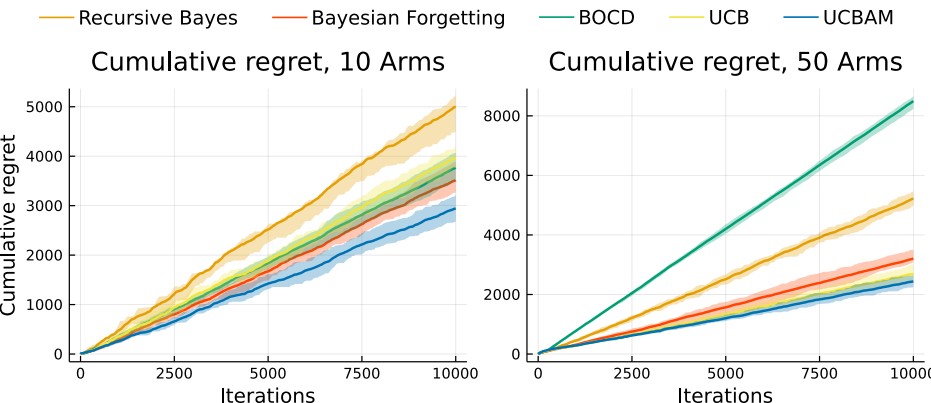

Figure 4: Cumulative regret over time between Thompson sampling (recursive Bayes), Bayesian exponential forgetting, BOCD, UCB, and UCBAM for different numbers of non-stationary arms, where results are averaged over 5 random arm configurations, where for each arm configuration results were collected over 5 different random seeds. Shaded bars represent 25-75% quantiles.

A common environment for testing the performance of online learning algorithms is the bandits setting (Sutton & Barto, 2018). We study a non-stationary version of the bandits setting where each arm switches between two values asynchronously, such that the best arm could be, at any point in time, a previously low value arm. Gaussian noise with $\sigma = 0.25$ is additionally added to the current arm value. Sample arm values can be found in Figure 5.

For stationary bandits, a popular algorithm is Thompson sampling (Thompson, 1933) in which a posterior over each arm is continually updated via recursive Bayes. These posteriors are then

leveraged to decide which arm the agent should pull, where the posterior uncertainty allows the agent to automatically switch between exploration and exploitation. In the non-stationary setting, we would expect vanilla Thompson sampling to fail as the arm posteriors would continue becoming more certain, as is evident from section 4.1. While there are many approaches for how to adapt BAM to perform well in the non-stationary bandits setting, we take a simple approach and combine BAM with the upper confidence bound (UCB) bandit algorithm (Agrawal, 1995), which we call UCBAM; in the interest of space, we provide an algorithm table in Appendix D.3.1. We compare UCBAM against UCB, Thompson sampling, Bayesian exponential forgetting + Thompson sampling and a BOCD + Thompson sampling scheme proposed by Mellor & Shapiro (2013); hyperparameter values can be found in Appendix D.3. From Figure 4, we see that UCBAM outperforms the other methods for both 10 and 50 arms. Thompson sampling fails to capture the true current values of the arms and suffers a large penalty while exploration afforded by UCB enables better performance. BOCD achieves low regret in the 10 arm setting, but reverts to its prior too often to perform well with 50 arms.

## 4.4 EXPERIMENT 4: DOMAIN ADAPTATION WITH ROTATED MNIST

In the image classification setting, we often want to operate across a variety of domains. Traditional approaches include learning a single high capacity model or encoding assumptions about the domain structure into the system (Jaderberg et al., 2015; Worrall et al., 2017). Instead, we use a simple multivariate linear regression model where the targets are one-hot encoded labels, taking the highest output as the selected class. We consider a setting where the distribution of domains is known and is the same at both train and test time and evaluate BAM's ability to classify given a small number of labeled examples from the domains to adapt its belief. To achieve this, we create a rotated MNIST dataset. 32 domains were created, where each domain was comprised of 1,875 randomly sampled without replacement from the training set. In a domain, the images are rotated by an angle sampled uniformly at random from 0 to $\pi$. Each domain is treated as one batch of data in the memory buffer, i.e. $\mathcal{D}_i = \{(x_j^i, y_j^i)\}_{j=1}^{1875}$. We split and rotate the test set similarly into 8 domains and give 10 labeled examples from each to find readout weights over the training data. We calculate the average accuracy over all test domains and collect results over 10 random seeds. While OLS trained over all domains get a mean and standard deviation accuracy of $55\% \pm 3.7\%$ accuracy, BAM is able to achieve a test set accuracy of **71.8% ± 5.2%**, showing that BAM is able to leverage previous experiences to adapt to novel domains.

## 5 CONCLUSION AND FUTURE WORK

In this work we present BAM, a flexible Bayesian framework that allows agents to adapt to non-stationary environments. Our key contribution is the addition of a memory buffer to the Bayesian framework, which allows the agent to adaptively change its prior by choosing which past experiences to remember and which to forget. Empirically, we show the proposed approach is general enough to be deployed in a variety of problem settings such as online inference, control, non-stationary bandits and domain adaptation. To ensure that we isolated the benefits of BAM, the experiments focused on conjugate-prior distributions as it allowed us to compute the prior/posterior and the log-marginal likelihood in closed form. Future work will focus on leveraging advances in streaming variational inference (Broderick et al., 2013; Kurle et al., 2020) to allow BAM to be deployed on more complicated models, i.e. Bayesian deep neural networks. For simplicity, we focused on binary values for the readout weights as it allowed for a simple greedy discrete optimization algorithm to be used. We imagine that allowing the weights to be any value between 0 and 1 will increase performance in certain settings and allow BAM to have a much larger repertoire of priors that it can construct, as well as suggest different optimization algorithms to use within the framework. Finally, efficient memory buffer schemes will be explored to avoid the 'infinite memory' problem of continual learning, enabling BAM to operate with efficiency indefinitely.

## 6 ACKNOWLEDGMENTS

The authors thank Ayesha Vermani, Matthew Dowling and Il Memming Park for insightful discussions and feedback.

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

## A  EXAMPLES OF POSTERIORS WITH DECREASING VARIANCE

In this section we will provide two cases where the variance of the posterior is non-increasing with probability 1 as more data is collected, regardless of the observed data. For simplicity we stick to only 1D, though we are confident these results extend to their multi-dimensional extensions.

### A.1  BAYESIAN ESTIMATION OF MEAN OF NORMAL DISTRIBUTION

The likelihood is of the form
$$p(y|\theta) = \mathcal{N}(\theta, \sigma^2), \tag{25}$$
where $\sigma^2 > 0$ is known. We use a normal prior
$$p(\theta) = \mathcal{N}(\bar{\theta}_0, \tau_0), \tag{26}$$
where $\tau_0 > 0$. Given arbitrary data $y_1, \cdots, y_N \sim p(y_{1:N})$ we get that the posterior is of the form
$$p(\theta|y_{1:N}) = \mathcal{N}(\bar{\theta}_N, \tau_N), \tag{27}$$
where
$$\tau_N = (\tau_0^{-1} + N\sigma^{-2})^{-1} = \frac{\sigma^2 \tau_0}{\sigma^2 + N\tau_0}, \tag{28}$$
$$\bar{\theta}_N = \tau_N \left( \tau_0^{-1}\bar{\theta}_0 + \sigma^{-2} \sum_{n=1}^{N} y_n \right). \tag{29}$$

We observe that the posterior variance, equation 28, is not a function of the observed data. In fact, the posterior variance is deterministic given $N$, $\tau_0$ and $\sigma^2$. In this particular setting, we can show that $\tau_N$ is a *strictly decreasing* function of $N$.

To prove that $\tau_0 > \tau_1 > \cdots > \tau_n > \cdots > \tau_N$, it suffices to show that
$$\tau_{n-1} > \tau_n, \quad \forall n \in \{1, \cdots N\}, \tag{30}$$
which is equivalent to showing that
$$\frac{\tau_n}{\tau_{n-1}} < 1, \quad \forall n \in \{1, \cdots N\}. \tag{31}$$

Before proceeding, we note that as Bayes' theorem is closed under recursion, we can always express the posterior variance as
$$\tau_n = (\tau_{n-1} + \sigma^{-2})^{-1} = \frac{\sigma^2 \tau_{n-1}}{\sigma^2 + \tau_{n-1}}. \tag{32}$$

Computing $\tau_n/\tau_{n-1}$

$$\frac{\tau_n}{\tau_{n-1}} = \frac{\sigma^2 \tau_{n-1}}{\sigma^2 + \tau_{n-1}} \times \frac{1}{\tau_{n-1}}, \tag{33}$$

$$= \frac{\sigma^2}{\sigma^2 + \tau_{n-1}}. \tag{34}$$

Because

$$\tau_n > 0, \quad \forall n \in \{0, \cdots, N\}, \tag{35}$$

we have that $\sigma^2 < \sigma^2 + \tau_{n-1}$, and conclude that $\tau_n/\tau_{n-1} < 1$.

### A.2 BAYESIAN LINEAR REGRESSION

Next, we consider the setting of Bayesian linear regression with known variance. The likelihood is of the form

$$p(y_i|x_i, \theta) = \mathcal{N}(\theta x_i, \sigma^2), \quad x_i \in \mathbb{R}, \tag{36}$$

where $\sigma^2 > 0$ is known. We use a normal prior

$$p(\theta) = \mathcal{N}(\bar{\theta}_0, \tau_0), \tag{37}$$

where $\tau_0 > 0$. Given arbitrary observations $(x_1, y_1), \ldots, (x_n, y_n)$, we have that the posterior is of the form

$$p(\theta|x_{1:N}, y_{1:N}) = \mathcal{N}(\bar{\theta}_N, \tau_N), \tag{38}$$

where

$$\tau_N = \left(\tau_0^{-1} + \sigma^{-2} \sum_{n=1}^{N} x_n^2\right)^{-1} = \frac{\sigma^2 \tau_0}{\sigma^2 + \tau_0 \sum_{n=1}^{N} x_n^2}, \tag{39}$$

$$\bar{\theta}_N = \tau_N(\tau_0^{-1}\bar{\theta}_0 + \sigma^{-2} \sum_{n=1}^{N} x_n y_n). \tag{40}$$

To prove that $\tau_0 \geq \tau_1 \geq \cdots \geq \tau_n \geq \cdots \geq \tau_N$, it suffices to show that

$$\frac{\tau_n}{\tau_{n-1}} \leq 1, \quad \forall x_n \in \mathbb{R}, \forall n \in \{1, \cdots, N\}. \tag{41}$$

Again, due to the Bayes being closed under recursion, we can always rewrite the posterior variance as

$$\tau_n = \left(\tau_{n-1}^{-1} + \sigma^{-2} x_n^2\right)^{-1} = \frac{\sigma^2 \tau_{n-1}}{\sigma^2 + \tau_{n-1} x_n^2}. \tag{42}$$

So

$$\frac{\tau_n}{\tau_{n-1}} = \frac{\sigma^2 \tau_{n-1}}{\sigma^2 + \tau_{n-1} x_n^2} \times \frac{1}{\tau_{n-1}}, \tag{43}$$

$$= \frac{\sigma^2}{\sigma^2 + \tau_{n-1} x_n^2}. \tag{44}$$

As $x_n^2 \geq 0$, we have that $\tau_n/\tau_{n-1} \leq 1$, which completes the proof.

## B PROOF OF PROPOSITION 1

For clarity, we rewrite the proposition below
**Proposition.** *Let*

$$p(\theta|\mathcal{D}_{<t}, W_t) \propto p(\theta) \prod_{j=1}^{t-1} p(\mathcal{D}_j|\theta)^{w_{t,j}}, \quad w_{t,j} \in \{0, 1\}, \tag{45}$$

*be the prior used in BAM and let*

$$p(\theta|\mathcal{D}_{<t}) \propto p(\theta) \prod_{j=1}^{t-1} p(\mathcal{D}_j|\theta), \tag{46}$$

*be the recursive Bayes prior. Then*

$$\mathbb{E}\left[Var(\theta|\mathcal{D}_{<t}, W_t)|W_t\right] \geq \mathbb{E}[Var(\theta|\mathcal{D}_{<t})], \quad \forall W_t \in \{0, 1\}^{t-1}. \tag{47}$$

*Proof.* We begin by describing some simple cases, before presenting the proof for the general case.

**Case 1: All the readout weights are 1.**
If all the readout weights are 1, i.e. $W_t = \mathbf{1}$ then

$$p(\theta|\mathcal{D}_{<t}, W_t = \mathbf{1}) = p(\theta|\mathcal{D}_{<t}), \tag{48}$$

recovering the recursive Bayes prior. Thus

$$\mathbb{E}\left[\mathrm{Var}(\theta|\mathcal{D}_{<t}, W_t = \mathbf{1})\big|W_t = \mathbf{1}\right] = \mathbb{E}[\mathrm{Var}(\theta|\mathcal{D}_{<t})]. \tag{49}$$

**Case 2: All the readout weights are 0.**
If all the readout weights are 0, i.e. $W_t = \mathbf{0}$ then

$$p(\theta|\mathcal{D}_{<t}, W_t = \mathbf{0}) = p(\theta), \tag{50}$$

recovering the base prior. The law of total variance states

$$\mathrm{Var}(\theta) = \mathbb{E}\left[\mathrm{Var}(\theta|\mathcal{D}_{<t})\right] + \mathrm{Var}(\mathbb{E}[\theta|\mathcal{D}_{<t}]). \tag{51}$$

As both terms on the right-hand side are positive, this implies that

$$\mathbb{E}\left[\mathrm{Var}(\theta|\mathcal{D}_{<t}, W_t = \mathbf{0})\big|W_t = \mathbf{0}\right] = \mathrm{Var}(\theta) \geq \mathbb{E}\left[\mathrm{Var}(\theta|\mathcal{D}_{<t})\right]. \tag{52}$$

**Case 3: General case**
Let $\mathbf{r}$ be the indices of the readout weight set to 1 ("remembered") and $\mathbf{f}$ be the indices of the readout weights set to 0 ("forgotten"). We can express the memory buffer as $\mathcal{D}_{<t} = \mathcal{D}_{\mathbf{r}} \cup \mathcal{D}_{\mathbf{f}}$ where $\mathcal{D}_{\mathbf{r}}$ are the data points selected by the readout weights and $\mathcal{D}_{\mathbf{f}}$ are the data points that are ignored. We can rewrite the BAM prior as

$$p(\theta|\mathcal{D}_{<t}, W_t) = p(\theta|\mathcal{D}_{\mathbf{r}}), \tag{53}$$

which is equivalent to applying Bayes theorem using $\mathcal{D}_{\mathbf{r}}$. Similarly, we can rewrite the recursive Bayes prior as

$$p(\theta|\mathcal{D}_{<t}) = p(\theta|\mathcal{D}_{\mathbf{r}}, \mathcal{D}_{\mathbf{f}}) \propto p(\mathcal{D}_{\mathbf{f}}|\theta)p(\theta|D_{\mathbf{r}}). \tag{54}$$

Using the law of total variance, we get

$$\mathrm{Var}(\theta|\mathcal{D}_{<t}, W) = \mathrm{Var}(\theta|\mathcal{D}_{\mathbf{r}}) = \mathbb{E}\left[\mathrm{Var}(\theta|\mathcal{D}_{<t})\big|\mathcal{D}_{\mathbf{r}}\right] + \mathrm{Var}\left(\mathbb{E}[\theta|\mathcal{D}_{<t}]\big|\mathcal{D}_{\mathbf{r}}\right), \tag{55}$$

where again, the above implies

$$\mathrm{Var}(\theta|\mathcal{D}_{\mathbf{r}}) \geq \mathbb{E}\left[\mathrm{Var}(\theta|\mathcal{D}_{<t})\big|\mathcal{D}_{\mathbf{r}}\right]. \tag{56}$$

As the above inequality holds for all values of $\mathcal{D}_{\mathbf{r}}$, it also holds under expectation as well

$$\mathbb{E}[\mathrm{Var}(\theta|\mathcal{D}_{\mathbf{r}})\big|W_t] \geq \mathbb{E}\left[\mathrm{Var}(\theta|\mathcal{D}_{<t})\big|W_t\right]. \tag{57}$$

Since $\mathrm{Var}(\theta|\mathcal{D}_{<t})$ is the variance under the recursive Bayes model, it is not a function of $W_t$, allowing the conditioning on $W_t$ to be dropped

$$\mathbb{E}\left[\mathrm{Var}(\theta|\mathcal{D}_{<t})\big|W_t\right] = \mathbb{E}\left[\mathrm{Var}(\theta|\mathcal{D}_{<t})\right]. \tag{58}$$

Applying our definition of $\mathcal{D}_{\mathbf{r}}$ recovers the desired result:

$$\mathbb{E}[\mathrm{Var}(\theta|\mathcal{D}_{<t}, W_t)\big|W_t] \geq \mathbb{E}\left[\mathrm{Var}(\theta|\mathcal{D}_{<t})\right]. \tag{59}$$

$\square$

## C  DISCUSSION OF GREEDY DISCRETE OPTIMIZATION

As the number of choices is $2^{(t-1)}$, it is impractical to use brute force methods for solving the discrete optimization problem defined in equation 21. For simplicity, we use two types of greedy approaches for discrete optimization. In both cases, each element in memory is evaluated against a target datum with the inner term of equation 19, the log marginal likelihood and regularization term. The first is a bottom-up approach, where we start with all readout weights set to 0 and greedily add the most beneficial associated datum until the combined score decreases. Pseudo code is displayed in Algorithm 1. Note that this is similar in spirit to the

stepwise selection approach used for selecting variables in linear regression (Hocking, 1976).

---

**Algorithm 1:** Bottom-Up Greedy for BAM

---

**Data:** memory $\mathcal{D}_{<t}$, target $\mathcal{D}_t$, prior $p$, regularizer strength $\lambda$
*priorscore* $\leftarrow \log p(\mathcal{D}_t)$ ;
**for** *size($\mathcal{D}_{<t}$)* **do**
    **for** *each $\mathcal{D}_i$ in $\mathcal{D}_{<t}$* **do**
        **if** *W[i] > 0* **then**
            | scores[i] $\leftarrow \log \int p(\mathcal{D}_t|\theta_t)p(\theta_t|W, \mathcal{D}_{<t})d\theta_t + \log p(W|\mathcal{D}_{<t})$
        **else**
            | scores[i] = -Inf
        **end**
    **end**
    *score, idx* = findmax(scores) ;
    **if** *score > priorscore* **then**
        $W[idx] \leftarrow 1$ ;
        *priorscore* $\leftarrow$ *score* ;
        $p$ = posterior($p, \mathcal{D}_{<t}[idx]$)
    **else**
        | return $W$
    **end**
**end**
**Result:** Readout weights $W$

---

In the second approach, the readout weight starts at 0. The contribution of each datum in $\mathcal{D}_{<t}$ is evaluated independently (and can be done practically in parallel with either multi-core CPUs or GPUs). These scores are filtered to only be scores better than the base prior's likelihood. The top $q\%$ percentile of remaining scores are chosen and their corresponding readout weight value are set to 1. Pseudo code is displayed in Algorithm 2. This approach is faster than bottom-up as only one round of optimization is needed but the combination of each of the individual experiences could potentially lead to sub-optimal performance. Additionally, the percentile cutoff may needlessly include or exclude weight values.

In practice, we found that the two approaches performed similarly with the main exception being the MNIST experiment, where the parallel approach was significantly worse than bottom-up.

---

**Algorithm 2:** Parallel selection for BAM

---

**Data:** memory $\mathcal{D}_{<t}$, target $\mathcal{D}_t$, regularizer strength $\lambda$, prior distribution $p$, cutoff $q$
*priorscore* = $\log p(\mathcal{D}_t)$ ;
**for** *each $\mathcal{D}_i$ in $\mathcal{D}_{<t}$* **do**
    | scores[i] $\leftarrow \log \int p(\mathcal{D}_t|\theta_t)p(\theta_t|\mathcal{D}_i)d\theta_t + \log p(W|\mathcal{D}_i)$
**end**
cutoff = quantile(*scores > priorscore, q*) ;
**for** *each in scores* **do**
    **if** *scores[i] > cutoff* **then**
        | $W[i] \leftarrow 1$
    **else**
        | $W[i] \leftarrow 0$
    **end**
**end**
**Result:** Readout weights $W$

---

# D    EXPERIMENTAL SETTINGS

## D.1    CONTROLS

For our controls experiments, we used Model Predictive Path Integral control (Williams et al., 2017), a model predictive control (MPC) algorithm with a planning horizon of 50 timesteps and 32 sample trajectories. Our sampling covariance was 0.4 for each controlled joint–in the case of Cartpole, the action space is 1. The temperature parameter we used was 0.5.

Planning with a probabilistic model involves each sampling trajectory to use a different model sampled from the current belief (as opposed to a sampled model per timestep); planning rollouts included noise, such that

$$x_t = x_{t-1} + M'\phi(x_{t-1}, a_t) + \varepsilon_t, \quad \varepsilon_t \sim \mathcal{N}(0, \sigma^2 I), \tag{60}$$

where $M'$ is sampled from the current belief. $\phi$ is the random Fourier features function from (Rahimi & Recht, 2007) where we use 200 features with a bandwidth calculated as the mean pairwise distance of the inputs (states and actions) which is 6.0. To learn $M$, we use Bayesian linear regression where

each row of $M$ is modeled as being independent. We place a multivariate Normal prior on each of the rows with a prior mean of all 0s and prior precision of $10^{-4}I$.

The Cartpole model's initial state distribution for positions and velocities were sampled uniformly from -0.05 to 0.05, with the angle of the cart being $\pi$ such that it points down. This sets up the swing-up problem.

For the episodic one-shot experiment, we perform MPC for 200 timesteps as one trial. 15 trials make one episode, with the dynamical properties of the environment (i.e. gravity) fixed for the duration of the trial. We vary the gravity parameter of the model by selecting gravity values from celestial bodies of the Solar System; we used Earth, Mars, and Neptune at 9.81, 3.72, and 11.15 $m/s^2$, respectively. At the start of a new episode, each method's beliefs are reset to the base prior, and each method proceeds to update their respective beliefs accordingly. BAM retains each trail's datum in memory across episodes.

For the continual learning experiment, we do not inform our agent that the model dynamics have changed, i.e. we never reset the agent's belief to a prior. Instead, we use Bayesian Online Changepoint Detection (BOCD) to discern if the underlying model distribution has changed. BOCD is compared against BAM, both with and without changepoint detection; while BOCD resets to a prior when a change is detected, BAM optimizes for a weight vector over the previously experienced data. The BOCD switching parameter $\lambda$ for its hazard function was set to 0.11. The agent attempts the task for 60 trials, with the environment experiencing changes 3 times during said trials.

## D.2 Domain Adaptation with Rotated MNIST

We ran 10 independent Bayesian linear regressions, one for each dimension of the one-hot encoded target. As the prior, we use a multivariate Normal distribution with a prior mean of all 0s and prior precision of $0.1I$. Similar to the controls experiment, we assume the additive noise is fixed and set to $\sigma^2 = 10^{-4}$. As regularization had little effect, we set $\lambda = 0$.

## D.3 Non-stationary Bandits

For both UCB and UCBAM, we use a confidence-level function of $f(t) = 1 + t\log^2(t)$. The timescale parameter for BOCD + Thompson sampling is 0.016, which is the expected frequency of the arm switches. The weighting term for Bayesian exponential forgetting + Thompson sampling is 0.8.

### D.3.1 Description of UCBAM

The challenge of bandit settings is the need to explore, especially in the non-stationary setting we devised. As such, UCB is a well known algorithm for leveraging the uncertainty in the arm values to enable exploration. We combine this frequentist method with BAM as follows. When we assume to 'know' the current best arm value, we exploit it and keep a belief over its distribution with BAM. The signal for whether the best arm is 'known' is if the likelihood of the current arm's value is higher with our current arm belief or higher with the naive base prior. If the base prior produces a higher likelihood, we assume the current arm distribution is incorrect (and will be updated with BAM), and we default to the UCB metric for arm selection. This simple combination of methods in this setting allows for the exploration benefits of UCB with the quick recognition of high value arms due to BAM and subsequent exploitation.

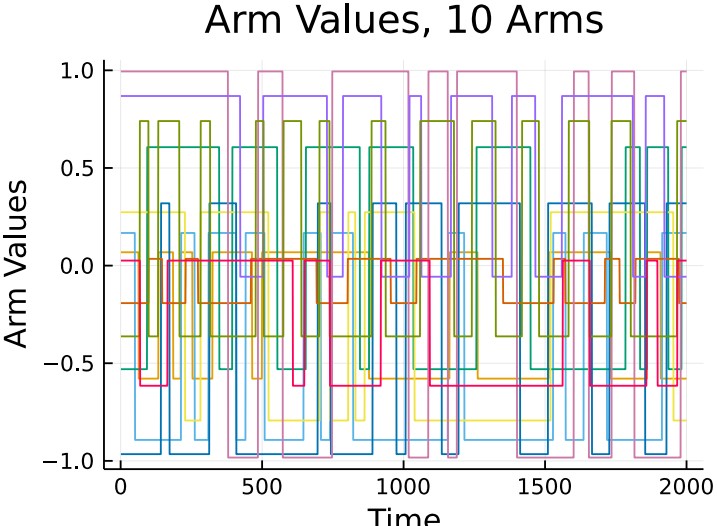

Figure 5: For reference, the arm values of the bandit experiments without added noise ($\sigma = 0.25$), with only 2000 time steps for clarity. Each arm switches between a high and low value at variable times, such that the highest value arm may be a previously low value arm. The simplest strategy would be to find the highest mean value arm, which is what UCB does in this case. Recursive Bayes attempts the same, but does not explore sufficiently to achieve an accurate estimate of the mean arm values.

---

**Algorithm 3:** UCBAM

---

**Data:** prior distribution $p$

$K \leftarrow$ number of arms ;

b = copy($p$), empty $\mathcal{D}$, K times ;  `# belief and memory per arm`

*known* $\leftarrow$ false ;

**for** *each iteration* **do**

    **if** *known* **then**

        | arm $\leftarrow$ thompson($b_{1...K}$)

    **else**

        | arm $\leftarrow$ UCB choice

    **end**

    $v \leftarrow$ pull(arm) ;

    **if** $log(p(v)) \geq log(b_{arm}(v))$ **then**

        | *known* $\leftarrow$ false

    **else**

        | *known* $\leftarrow$ true

    **end**

    $b_{arm} = \text{BAM}(p, \mathcal{D}^{arm}_{<t}, v)$ ;        `# BAM posterior update`

    $\mathcal{D}_{<t} = [\mathcal{D}_{<t}, v]$ ;        `# add value to memory`

**end**

---

