# OpenReview forum: "BAM: Bayes with Adaptive Memory"
_ICLR.cc/2022/Conference — ICLR 2022 Poster_

### Official Review · Reviewer_o4TK · 2021-10-30

**Correctness:** 4
**Technical Novelty And Significance:** 2
**Empirical Novelty And Significance:** 2
**Recommendation:** 6
**Confidence:** 4

**Main Review:**

The paper is well written and easy to follow. The motivation is clear and the problem is well set. The proposed learning framework is a reasonable solution in principle. The experiments clearly demonstrate the benefit of BAM, especially its ability to adapt to the change of environments.

However, I'm a bit skeptical about the practicability of the proposed framework. All the elements of the algorithm, including the posterior updates and greedy selection procedure, assume the traceability of the marginal likelihood, which is not true in most non-trivial real-world applications.  As the authors stated in the conclusion, one can introduce variational approximations or MCMC to conduct approximate inference, but these are way harder than it looks since now each evaluation of $p(\theta_t|W, D_{<t})$ requires an iterative procedure requiring heavy computation until convergence, so the greedy selection procedure can be prohibitively time-consuming. As the authors pointed out when relating BAM to the existing works, and also suggested as future work, one can allow the selection variables $W$ to be any real number between $[0,1]$. Then we can resort to gradient-based approximate optimization techniques (e.g., based on variational approximation with Gumbel-softmax), but at this time it is not clear how accurate all such approximate inference techniques (especially for high-dimensional models such as deep neural networks applied to large-scale data). One should also think about an alternative prior $p(W|\theta_t)$ because the KL-divergence needed for the current prior is intractable.

So my point is, there are many obstacles to be addressed if we are to extend the current BAM framework for realistic scenarios, and all of such issues pose their own research problems that might require contributions significant enough to write standalone papers. Therefore, I think the current submission is quite incomplete, although I agree that the research direction itself is interesting.

Another practical aspect that is worth pondering is how the proposed framework compares to existing continual learning techniques introduced for deep neural networks. There are plenty of works for deep continual learning, where the goal is to learn large-scale deep neural networks while taking a continuous stream of non-stationary task data. Existing continual learning techniques learn to adaptively forget or retain past experiences, and also try to minimize the cost of learning and the complexity of the model (minimize the unnecessary expansion of the model).  Especially, an important consideration in continual learning is that it typically assumes that an agent doesn't have access to the previous data ($D_{<t}$), but only to the model learned from it. This is a reasonable assumption because keeping all the past data requires huge memory. So the difficulty in continual learning comes from the fact that it should learn continuously adapting model which keeps a balance between 1) how much to forget and 2) how much to retain past experiences, without access to the previous data. For instance, variational continual learning (Nguyen et al., 2018) constructs a lightweight summary of previous data that can be used as a representative for the subsequent learning. As far as I can see, BAM assumes access to all the previous data whenever updating the posteriors, and this again can be a significant challenge for more realistic learning scenarios.

I'm also quite confused with the name "Bayes augmented with memory". What does the "memory" stand for? If it is for the past data being used for the update of the posteriors, isn't the vanilla recursive Bayes also augmented with memory? The difference between BAM and the recursive Bayes is in BAM's use of the readout variable $W$, but I don't think the variable $W$ itself stands for the "memory".

References
(Nguyen et al., 2018) Nguyen, C. V., Li, Y., Bui, T. D., and Turner, R. E. Variational continual learning. ICLR, 2018.

**Summary Of The Paper:**

This paper proposes a learning framework called Bayes Augmented with Memory (BAM), where a recursive adaptation of the Bayes formula is augmented with selection variables $W$ to allow an agent to adaptively choose past experiences to forget. The original recursive Bayes formula assumes the stationarity of the data generating distribution, so keep all the past data for updating the posteriors for newly arriving data, and thus always tends to decrease the variance of the posteriors as it is meant to be. However, this can severely fail when the stationarity assumption is violated; when an agent encounters a change in the environment, it should somehow discard the past experience and quickly adapt to new data. BAM explicitly models this procedure with the binary selection variables, and by greedily optimizing those selection variables at each entrance of new data, quickly adapt to the change of environment and still leverages the past experiences. Various experiments with non-stationary environments demonstrate the usefulness of the proposed framework.

**Summary Of The Review:**

I think the proposed framework can be potentially interesting and useful, but at current presentation lacks many practical considerations to be dealt with, especially how it can be applied to non-trivial models requiring approximate inference for computing posteriors. It is quite disappointing thus to only see experiments on conjugate models.

---

> ### Author Response · Authors · 2021-11-16
> **Response to reviewer**
>
> We thank the reviewer for their feedback. The reviewer mentioned several extensions to BAM that would improve its practical applicability, such as techniques for approximating the posterior for streaming data [1, 2, 3, 4] and techniques for selecting which data points to keep in memory [1, 2, 4]. We agree that there are many exciting techniques that have been explored in the online/continual learning literature to address scalability. However, we would like to emphasize the placement of this paper in the literature, and further expand upon what we stated in our general response. We intentionally kept the presentation of BAM lightweight in order to emphasize the key idea (selectively remembering and forgetting past data based on recently observed data), and clearly demonstrate its benefit through the experiments as the reviewer noted, without conflating this key idea with implementation details and externalities. Different real-world problems may suggest or require different optimization algorithms or memory buffer methods within the overarching BAM framework. We agree with the reviewer that many of these implementation details warrant their own papers and lines of research, and we believe that many techniques from related work [1, 2, 3, 4] can be fruitfully combined with BAM.
>
> Concerning continual learning techniques for deep neural networks, we first note that the continual learning paradigm assumes an i.i.d setting, which allows the use of naive recursive Bayes in works such as [1]. Thus, the focus of these works is not a new framework, but on extensions to recursive Bayes in the streaming data case [1, 2, 3]. Moreover, for methods that explicitly deal with Bayesian neural networks in the non-stationary case i.e. [2, 3], they use techniques such as exponential forgetting [2]--which is generalized by BAM--or assuming an AR process on the weights of the neural network [3]. Also, in [1, 2] selective data points are remembered from the past to help get a better variational approximation. We view these techniques for better online variational inference and minimizing memory as complementary to BAM, and we believe these techniques could be leveraged to extend BAM to more complicated problems.
>
> To address the name and our use of memory in the BAM framework, “memory” refers to the (re-)use of specific data points previously encountered. While recursive Bayes also incorporates past data, the memory is implicit in the informed prior, whereas BAM has an explicit memory that can be accessed repeatedly, allowing the agent to adaptively choose which memories to remember and which to forget. This adaptivity is distinct when compared to recursive Bayes or Bayesian Forgetting, in that what is implicitly remembered in those methods is fixed. We hope that our use of the term “memory” reminds the reader of the selective nature of BAM’s posterior.
>
> [1] Variational Continual Learning. Nguyen et al., ICLR 2018.
> [2] Continual Learning with Bayesian Neural Networks for Non-Stationary Data. Kurle et al., ICLR 2020.
> [3] Dynamic Bayesian Neural Networks. Rimella & Whiteley, arXiv 2020.
> [4] Streaming Variational Bayes. Broderick et al., NeurIPS 2013.

---

> > ### Comment · Reviewer_o4TK · 2021-11-22
> > **Response**
> >
> > Thanks for the clarification.
> >
> > Unfortunately, the response still doesn't resolve my initial concerns on the paper.
> >
> > 1) Practicality of the algorithm:  I think the proposed framework is interesting, but for a top conference like ICLR, one would need more than a theoretically attractive framework with some showcase experiments on easy cases.  I'm not saying this because the algorithm is not applied for deep neural networks with large-scale data; it was not even validated on non-conjugate cases for which the computation of posteriors and marginals are interactive, so the proposed algorithm (especially the greedy selection part) is not warranted to work efficiently as showcased in this paper. If the paper was to be accepted solely based on the concept and synthetic experiments, the framework itself should be completely new and original, but I don't think BAM is the case.
> >
> > 2) Continual learning: we first note that the continual learning paradigm assumes an i.i.d setting <- I respectively disagree with this. Continual learning does assume non-stationary data.  For instance, see this sentence I brought from (Parisi et al., 2019), " However, lifelong learning remains a long-standing challenge for machine learning and neural network models since the continual acquisition of incrementally available information from non-stationary data distributions generally leads to catastrophic forgetting or interference.". One of the main challenges of continual learning (a.k.a. lifelong learning) is to prevent catastrophic forgetting during the model adaptation whenever the model encounters changes in data distribution. Many continual learning papers (Yoon et al 2018, Li et al., 2019) test their models by the stream of data with changes in the data generating processes (e.g., gradually permuting MNIST images, using different splits of CIFAR images, ..).  I'm pretty sure that a naive recursive Bayes would fail for the typical benchmarks being used in continual learning literature, and the proposed method is worth comparing to some baselines in continual learning, especially the ones that do not need to keep all the past data.
> >
> >
> > Parisi et al., Continual lifelong learning with neural networks: A review, Neural Networks, 2019.
> > Yoon et al., Lifelong learning with dynamically expandable networks, ICLR 2018
> > Li et al., 2019, Learn to Grow: A Continual Structure Learning Framework for Overcoming Catastrophic Forgetting, ICML 2019.

---

> > > ### Author Response · Authors · 2021-11-22
> > > **Response**
> > >
> > > 1. If there is a reference that is similar to the framework we present that detracts from originality, then we would appreciate knowing what it is. As it stands, BAM is a general framework that is, to the best of our knowledge, new and original. Concerning our experiments, BAM makes no assumption on the prior/posterior nor on the problem formulation, merely requiring posterior calculations and some memory. Specific problems require specific solutions for posterior approximation and memory; this is evident from the first three references we cited in our previous response that each uses different variational approximations. BAM, just like any framework, would need to be adapted for a target domain and, as you mentioned, domain-specific implementations of BAM would be their own standalone papers. We do appreciate the need to demonstrate results on intractable posteriors but as we mentioned previously, the focus of this paper was the presentation of the framework and as such, we wanted to isolate the benefits of only BAM and not have it be compounded with things such as approximation schemes. Moreover, our experiments cover a wide range of domains: from model-based control to bandits to domain adaptation. Even if all the experiments only deal with conjugate priors, we believe that our breadth of experiments is more than enough empirical evidence about the benefits of BAM.
> > > 2. We respectfully disagree with the reviewer about their second comment. We first note that in many continual learning setups the goal is to prevent catastrophic forgetting. In contrast, when dealing with non-stationary data it is important to forget outdated information, as was stated in [1]  “Consequently, (continual learning) CL systems for non-stationary data require adaptation methods, which deliberately forget outdated information.” We do note that there are some nuances between the references the reviewer provided and the references we provided. The references cited by the reviewer [5, 6], are non-probabilistic and are engineered specifically for neural networks; thus, a comparison to these methods is not directly applicable.
> > > In contrast, the references we cited [1, 2, 3] and BAM are both probabilistic, specifically Bayesian. We want to emphasize that most Bayesian approaches for continual learning rely on naive recursive Bayes. For instance, variational continual learning (VCL) [1] is based on naive recursive Bayes, which is the very first equation in [1]. This point was made again in [2] “For instance, VCL [Nguyen et al., 2018] is a popular framework that uses a model’s previous posterior distribution as the prior for new data. However, the assumption of such continual learning setups is usually that the data distribution is stationary and not subject to change, in which case adaptation is not an issue.” As such, the novelty of the Bayesian continual learning approaches is in the variational inference scheme. For dealing with non-stationary data, most approaches just use techniques like Bayesian forgetting [2], BOCD [4], or do Bayesian filtering by assuming an AR(1) process on parameters [3]. As we have compared to BOCD and Bayesian filtering in the manuscript, we have compared to continual learning baselines and demonstrated the value of our approach.
> > >
> > > [1] Variational Continual Learning. Nguyen et al., ICLR 2018.
> > > [2] Continual Learning with Bayesian Neural Networks for Non-Stationary Data. ICLR 2020.
> > > [3] Dynamic Bayesian Neural Networks. arXiv 2020.
> > > [4] Detecting and Adapting to Irregular Distribution Shifts in Bayesian Online Learning. Li et al., NeurIPS 2021.
> > > [5] Lifelong learning with dynamically expandable networks. Yoon et al., ICLR 2018.
> > > [6] Learn to Grow: A Continual Structure Learning Framework for Overcoming Catastrophic Forgetting. Li et al., ICML 2019.

---

> > > > ### Comment · Reviewer_o4TK · 2021-11-23
> > > > **Response**
> > > >
> > > > Thanks for the further clarification.
> > > >
> > > > 1. Well, for me, the proposed framework, although well-executed, and conceptually attractive, is not that really surprising. I'm not saying that I have actually seen similar previous works before. My point is, I would have expected more in order for a conceptual framework without a practical demonstration to be accepted. But I admit that this is completely my own personal standard, so I cannot argue to reject the paper due to my personal standard.
> > > >
> > > > 2. Even if I still think the proposed work should be compared to the continual learning methods, I get the point of the authors casting the proposed framework as a Bayesian inference method. After all, you need to develop an approximate version of the proposed algorithm in order to compare to those continual learning baselines. So it again boils down to the problem of impracticality.
> > > >
> > > > Summing up, I'm still standing on my initial point; the proposed algorithm is not practical, since nothing has been shown for non-trivial models without conjugacy.  Considering the discussions with authors and clarifications, I raise my score to 6. I recommend discussing more about the practical considerations when one is to apply BAM to non-conjugate cases (especially how the greedy selection procedure can be made efficient even for intractable marginal likelihoods), and add some discussions on the continual learning literature.

---

### Official Review · Reviewer_EUxk · 2021-11-01

**Correctness:** 3
**Technical Novelty And Significance:** 3
**Empirical Novelty And Significance:** 2
**Recommendation:** 5
**Confidence:** 4

**Main Review:**

The paper tackles an important problem of Bayesian online learning in a non-stationary environment. The main contribution is BAM, an algorithm that constructs an informative prior distribution for current observations based on stored previous data. While the method seems technically sound, the method still falls short of the following points:

1) **Scalability**: the algorithm remembers all the data so far, which requires infinite memory in an ever-running system. So as to the computation. The algorithm will fail in an online learning environment. An immediate improvement would be to use a fixed memory. A similar idea occurs in continual learning with a memory system where people only select representative data into a fixed memory (see the following point).

2) **Proper related work discussion**: At least there are two highly-related papers not mentioned in the current version:

    - Li, Aodong, et al., “Detecting and Adapting to Irregular Distribution Shifts in Bayesian Online Learning”, NeurIPS 2021 (previously occurred in workshops).
    - Kurle, Richard, et al., "Continual learning with bayesian neural networks for non-stationary data." ICLR 2020.

    *(Possibly) related work in other fields*:

    *Continual learning with a fixed-size memory*:
    - Aljundi, Rahaf, et al. "Gradient-based sample selection for online continual learning." Advances in Neural Information Processing Systems 32 (2019): 11816-11825.

    *Bayesian inference with weighted likelihood*:
    - Wang, Yixin, Alp Kucukelbir, and David M. Blei. "Robust probabilistic modeling with bayesian data reweighting." International Conference on Machine Learning. PMLR, 2017.
    - Mandt, Stephan, et al. "Variational tempering." Artificial Intelligence and Statistics. PMLR, 2016.

3) **Large-scale experiments**: Current experiments are all analytical. However, practical systems could be complex, intractable, and large-scale in time. Without demonstrating the applicability in these environments, the approach may be not convincing.

4) **Baseline comparisons**: The paper conducted thorough experiments with baselines including ordinary Bayesian online learning and BOCD. However, existing baselines have obvious limitations and couldn’t properly illustrate the benefits of the additional memory and re-visited states. More adaptive baselines like Bayesian Forgetting or Bayesian filter can be desirable to exemplify the usefulness of memory in a changing environment.

Other additional comments/suggestions:
* $p(W_t|D_{<t}) \propto 1$: Should $W_t$ take some value?
* May consider change colors in Fig. 1 (left). Hard to distinguish methods.
* Fig. 5 is out of the main text.


**Summary Of The Paper:**

The paper aims to solve the downside of the posterior shrinkage of Bayesian online learning when applied in a non-stationary environment. The previously posterior (a.k.a., the current prior) could be misspecified for current observations with the posterior shrinkage due to the environment change. To solve this misspecification problem, the proposed method constructs an adaptive prior that is correctly specified for current observations. This adaptive prior distribution is constructed by selecting the previous relevant data samples, which requires storing the whole history. The method is evaluated on various analytical experiments.

**Summary Of The Review:**

The paper could be an interesting contribution in the area of Bayesian online learning in a non-stationary environment. However, unless solving the scalability problem and demonstrating the benefits of the memory, the paper isn’t persuasive enough.

---

> ### Author Response · Authors · 2021-11-16
> **Response to reviewer**
>
> We thank the reviewer for their feedback. The challenge of avoiding infinite memory is an active area of research. We do agree with the reviewer that in the general case, BAM needs to remember a representation for every batch of data--either through sufficient statistics or approaches used in [1,3]--which would require infinite memory in the infinite horizon case. There are methods available in the literature to operate within a fixed memory buffer i.e. [4], and we believe any of these could be adapted to work with BAM. The best method will undoubtedly depend on the problem at hand, and in our experimental evaluation, we wanted to decouple the benefits of BAM from the performance of individual buffering methods.
>
> We like the idea of comparing against Bayesian Forgetting, which BAM generalizes because such a comparison will demonstrate the value of data-adaptive forgetting. We will include this comparison soon.
>
> Concerning the lack of large-scale experiments, see our response to reviewer o4TK.
>
> We thank the reviewer for bringing these works to our attention. We have updated our manuscript accordingly, but we will also discuss these works now. While the method presented in [2] is similar in spirit to BAM there are very important differences between the two. First, the approach proposed in [2] is designed specifically for Gaussian priors/posteriors while BAM is designed for arbitrary posteriors (as was demonstrated in experiment 1 where we dealt with count-based data) and we have also proved in Theorem 1 that our approach allows for BAM to modulate its uncertainty relative to recursive Bayes, regardless of the form of the posterior. Second, in [2] when a shift in the environment is detected, the variance of the prior is increased by a *fixed amount*--which is chosen a priori--while keeping the mean fixed. In contrast, the use of readout weights allows for BAM to adaptively change its uncertainty.
>
> While [3] deals with Bayesian neural networks for non-stationary data, we note that the main contribution of the paper is an online variational inference scheme that can be combined with BAM. Moreover, to make the approach amenable for non-stationary data, the authors use exponential forgetting which is a subset of BAM.
>
> We thank the reviewer for pointing out [4] as it can be useful for allowing BAM to scale to more complicated models.
>
> Lastly, the use of readout weights may be seen as an instance of tempering the likelihood which has been proposed in [5,6]. We do stress that in [5,6] they deal with the offline case where data has already been collected. In contrast, BAM is designed for the online case.
>
> [1] Variational Continual Learning. Nguyen et al., ICLR 2018.
> [2] Detecting and Adapting to Irregular Distribution Shifts in Bayesian Online Learning. Li et al., NeurIPS 2021.
> [3] Continual Learning with Bayesian Neural Networks for Non-Stationary Data. Kurle et al., ICLR 2020.
> [4] Gradient based sample selection for online continual learning. Aljund et al., NeurIPS 2019.
> [5] Robust Probabilistic Modeling with Bayesian Data Reweighting. Wang et al., ICML 2017.
> [6] Variational Tempering. Mandt et al., ICML 2016.

---

> > ### Author Response · Authors · 2021-11-22
> > **Update**
> >
> > We wanted to inform the reviewer that we have added comparisons to Bayesian forgetting for the experiments in sections 4.1 and 4.3.

---

> > > ### Comment · Reviewer_EUxk · 2021-11-22
> > > **Thanks for the response**
> > >
> > > Thanks for the author's efforts in adding up new comparisons against the Bayesian forgetting baseline. The new experiments make the paper stronger than before, especially in demonstrating the benefits of a memory module. If Bayesian forgetting also applies to other experiments, please apply it to them as well.
> > >
> > > However, my concern for the algorithm's scalability still exists. I acknowledge the present paper converses technical statements of BAM and showcases its applicability in a class of problems. The author also mentioned their algorithm's compatibility with other fixed-memory strategies. I respect the theoretical thinking. But without the concrete algorithmic prototypes and concrete experiments, it is hard to justify the correctness of the statement. Besides, the scalability of intractable models, in my opinion, is another important indicator for the applicability in practical setups.

---

> > > > ### Author Response · Authors · 2021-11-22
> > > > **Response**
> > > >
> > > > While we believe the BAM framework to be general, we do not assume it automatically solves all problems, and experimental validation is necessary. The inputs and methods within the framework, (i.e. the memory buffer and posterior calculations), change on a per problem basis. In the experimental problems in this paper, BAM does not assume anything about the problem setups.
> > > >
> > > > If there is a specific problem setting that the reviewer is concerned about, we would appreciate knowing which such that we can discuss specifics, otherwise we can only address scalability in the general case. In the general case, BAM is a framework that uses memory, not one that selects how the memory buffer is formed. It operates on the memory buffer applied to it and attempts to provide the best memory-augmented posterior possible with said buffer. In its current formulation, BAM makes no attempt to select or structure what data is stored in the memory buffer, and it is because of this reason we assert that BAM is compatible with other memory strategies: the user can select the strategy most appropriate for their problem setting without changing how BAM operates. There is a strict separation that contributes to BAM's modularity.
> > > >
> > > > If the reviewer would prefer to see BAM applied to different memory strategies, we agree that this would be interesting work but out of scope for this paper presentation. Hypothetically, if BAM is held fixed with different memory strategies, and the experiment demonstrated some positive and some negative results, then the cause would be the memory strategies, not BAM. If instead, you are requesting that BAM provide a memory strategy, then we again assert that BAM does not solve the problem of perfect memory buffer formation as that is well known to be problem-specific.
> > > >
> > > > Similarly, with respect to intractable models, BAM may work for some methods of approximating intractable prior/posteriors and may not for others. The BAM framework itself would have remained unchanged. Methods of approximating intractable models are also orthogonal work that we do not attempt to solve.

---

### Official Review · Reviewer_1Svc · 2021-11-01

**Correctness:** 4
**Technical Novelty And Significance:** 4
**Empirical Novelty And Significance:** 4
**Recommendation:** 8
**Confidence:** 3

**Main Review:**

In this paper, the authors propose a new framework, Bayes Augmented with Memory (BAM), that takes advantage of past experience by allowing the agent to choose which past observations to remember and which to forget and demonstrate that BAM generalizes many popular Bayesian update rules for non-stationary environments. The variety of experiments demonstrate the ability of BAM to continuously adapt in an ever-changing world.

To the best of my knowledge, this is generally a good paper with a clear central idea. I have only two minor concerns:

For simplicity, this paper focused on binary values for the readout weights as it allowed for a simple greedy discrete optimization algorithm to be used. This assumption limits its practical application scenarios.

Although a simple greedy discrete optimization algorithm to be used, it still makes the proposed BAM have a relatively high time complexity. The paper could be improved if the authors can provide the time complexity of the proposed BAM. If it is difficulty, the comparison results in terms of the running time are needed.


**Summary Of The Paper:**

In this paper, the authors propose a new framework, Bayes Augmented with Memory (BAM), that takes advantage of past experience by allowing the agent to choose which past observations to remember and which to forget and demonstrate that BAM generalizes many popular Bayesian update rules for non-stationary environments.

**Summary Of The Review:**

In this paper, the authors propose a new framework, Bayes Augmented with Memory (BAM), that takes advantage of past experience by allowing the agent to choose which past observations to remember and which to forget and demonstrate that BAM generalizes many popular Bayesian update rules for non-stationary environments. The variety of experiments demonstrate the ability of BAM to continuously adapt in an ever-changing world.

To the best of my knowledge, this is generally a good paper with a clear central idea. I have only two minor concerns:

For simplicity, this paper focused on binary values for the readout weights as it allowed for a simple greedy discrete optimization algorithm to be used. This assumption limits its practical application scenarios.

Although a simple greedy discrete optimization algorithm to be used, it still makes the proposed BAM have a relatively high time complexity. The paper could be improved if the authors can provide the time complexity of the proposed BAM. If it is difficulty, the comparison results in terms of the running time are needed.

---

> ### Author Response · Authors · 2021-11-16
> **Response to reviewer**
>
> We thank the reviewer for their comments and appreciate their concern about algorithmic time complexity and binary weights. For buffer size N, the worst-case time complexity of the greedy and bottom-up algorithms presented in this work are $\mathcal{O}(N)$ and $\mathcal{O}(N^2)$, respectively; we have updated the manuscript accordingly. As noted in the general response, extensions exist for more efficient algorithms if they are needed. Implicit in the reviewer’s query about binary vs non-binary weights is the assumption that practical applications require fractional weights. Fractional weights are used in Bayesian Forgetting, where past observations are down-weighted according to a fixed exponential schedule. However, in the more flexible framework of BAM (which generalizes Bayesian Forgetting), it remains for future work to determine when fractional weights provide an advantage over 0/1 weights.

---

### Official Review · Reviewer_4Gvr · 2021-11-03

**Correctness:** 4
**Technical Novelty And Significance:** 4
**Empirical Novelty And Significance:** Not applicable
**Recommendation:** 8
**Confidence:** 3

**Main Review:**

Strengths:

- The theoretical development is principled;

- This approach seems to be novel (though I'm not familiar with the literature on Bayesian methods for non-stationary environments);

- The paper is clearly written.

Concerns:

None.

Minor points:

- In equations (11) and (12), shouldn't $\theta_t$ be $\theta$, as it is assumed not to change with $t$ here?

- Equation (17) is just a normalisation constant. Since (14) and (16) are also given only up to constants, is it worthwhile to include?

- Several times in section 2.1: "preventing BAM from overfitting" sounds like BAM won't overfit at all. Could you change the wording to reflect that regularisation reduces the chance/severity of overfitting, without preventing it altogether?

- Below equation (23) and in appendix C: $(t-1)!$ should be $2^{t-1}$

- Algorithm 1: While "Bottom's Up" [sic] is a nice name, I think the term you're looking for is "Bottom-Up" :)

**Summary Of The Paper:**

In this paper, a method BAM is proposed for Bayesian learning in non-stationary environments: basically, at each time step, each previous datum may or may not be incorporated into the new posterior, so that old data from different states can be ignored, while old data from the same or similar states is remembered. The method doesn't rely on parametric assumptions. Experiments demonstrate it to work well in various scenarios.

**Summary Of The Review:**

I enjoyed reading this paper. Assuming that this idea is as novel as claimed, I think this is a valuable contribution to the community and recommend acceptance.

---

> ### Author Response · Authors · 2021-11-16
> **Response to reviewer**
>
> We thank the reviewer for their positive feedback and for pointing out the typos; we have updated the manuscript to reflect these changes and in addition, we will respond to specific comments below.
>
> 1. “In equations (11) and (12), shouldn't $\theta_t$ be $\theta$, as it is assumed not to change with t here?” We use $\theta_t$ to keep the notation consistent across the paper but we note that in footnote 1, we demonstrate that we can still recover recursive Bayes using this formulation.
> 2. “Equation (17) is just a normalization constant. Since (14) and (16) are also given only up to constants, is it worthwhile to include?” We note that equation 17 is pivotal to how BAM selects the readout weights so we thought it pertinent to display.
> 3. “Several times in section 2.1: "preventing BAM from overfitting" sounds like BAM won't overfit at all. Could you change the wording to reflect that regularisation reduces the chance/severity of overfitting, without preventing it altogether?” We agree and have updated the manuscript accordingly.

---

### Author Response · Authors · 2021-11-16
**Response to all reviewers**

We’d like to thank the reviewers for spending the time to send us quality notes and their thoughts on this work. The reviewers’ acknowledgment of the challenges of the non-stationary problem setting and the value of our Bayesian framework BAM is appreciated. We will address individual reviewers’ concerns directly and wish to first address some common notes provided by the reviews, mainly regarding ‘scale’ found in this particular work.

We first want to reemphasize that the goal of this work is to introduce a novel framework for Bayesian online learning in nonstationary environments titled BAM. Multiple reviewers felt that while the framework is sound and the experiments demonstrate validity, more could be done to address ‘scale’ in terms of memory use and algorithm complexity. In short, we agree that scaling to interesting real-world problems would require careful consideration of methods and modifications. As BAM is a general and flexible framework, one can leverage methods that address specific goals like efficient data buffer use [3] and online approximation of the posterior and log marginal likelihood [1, 2]. Our goal in this work was to produce a clear and understandable presentation of this framework that selectively uses a memory of past data. To ensure that we isolate the benefits of BAM, we focused on conjugate priors which remove the effect of posterior approximations and alternative memory consolidation schemes. We acknowledge that details addressing specific problem constraints are interesting, but potentially orthogonal research topics. As reviewer o4TK noted, subsequent works that address these particulars would be their own stand-alone papers.

[1] Variational Continual Learning. Nguyen et al., ICLR 2018.
[2] Continual Learning with Bayesian Neural Networks for Non-Stationary Data. Kurle et al., ICLR 2020.
[3] Gradient based sample selection for online continual learning. Aljund et al., NeurIPS 2019.

---

### Decision · Program_Chairs · 2022-01-20

**Decision:**

Accept (Poster)

**Comment:**

The article introduces a Bayesian approach for online learning in non-stationary environments. The approach, which bears similarities with weighted likelihood estimation methods, associate a binary weight to each past observation, indicating if this observation should be including or not to compute the posterior. The weights are estimated via maximum a posteriori.

The paper is well written, the approach is novel and its usefulness demonstrated on a number of different experiments. The original submission missed some relevant references that have been added in the revision. The approach has some limitations, highlighted by the reviewers:
* it requires to solve a binary optimisation problem whose complexity scales exponentially with the size of the dataset; although the greedy procedure proposed by the authors seems to work fine on the examples shown, the approach may not be applicable to larger datasets
* it requires to store all the data
* it requires the traceability of the marginal likelihood

Despite these limitations, there was a general agreement that this paper offers a novel and useful contribution, and I recommend acceptance.

As noted by reviewer o4TK, I also think that the title is not very accurate. Bayesian methods naturally allow recursive updates of one's beliefs, and therefore have "memory". Maybe change the title for "Bayes with augmented selective/adaptive memory"?